# The Effect of Natural Antioxidants in the Development of Metabolic Syndrome: Focus on Bergamot Polyphenolic Fraction

**DOI:** 10.3390/nu12051504

**Published:** 2020-05-21

**Authors:** Cristina Carresi, Micaela Gliozzi, Vincenzo Musolino, Miriam Scicchitano, Federica Scarano, Francesca Bosco, Saverio Nucera, Jessica Maiuolo, Roberta Macrì, Stefano Ruga, Francesca Oppedisano, Maria Caterina Zito, Lorenza Guarnieri, Rocco Mollace, Annamaria Tavernese, Ernesto Palma, Ezio Bombardelli, Massimo Fini, Vincenzo Mollace

**Affiliations:** 1Institute of Research for Food Safety & Health IRC-FSH, University Magna Graecia, 88100 Catanzaro, Italy; micaela.gliozzi@gmail.com (M.G.); xabaras3@hotmail.com (V.M.); miriam.scicchitano@hotmail.it (M.S.); federicascar87@gmail.com (F.S.); boscofrancesca.bf@libero.it (F.B.); saverio.nucera@hotmail.it (S.N.); jessicamaiuolo@virgilio.it (J.M.); robertamacri85@gmail.com (Roberta Macrì); rugast1@gmail.com (S.R.); oppedisanof@libero.it (F.O.); mariacaterina.zito@studenti.unicz.it (M.C.Z.); lorenzacz808@gmail.com (L.G.); rocco.mollace@gmail.com (Rocco Mollace); an.tavernese@gmail.com (A.T.); palma@unicz.it (E.P.); ezio.bombardelli@plantexresearch.it (E.B.); mollace@libero.it (V.M.); 2Department of Medicine, Chair of Cardiology, University of Rome Tor Vergata, 00133 Roma, Italy; 3Nutramed S.c.a.r.l., Complesso Ninì Barbieri, Roccelletta di Borgia, 88021 Catanzaro, Italy; massimo.fini@sanraffaele.it; 4IRCCS San Raffaele Pisana, 00163 Roma, Italy

**Keywords:** metabolic syndrome, plant extracts, natural antioxidant, polyphenols, bergamot

## Abstract

Metabolic syndrome (MetS) represents a set of clinical findings that include visceral adiposity, insulin-resistance, high triglycerides (TG), low high-density lipoprotein cholesterol (HDL-C) levels and hypertension, which is linked to an increased risk of developing type 2 diabetes mellitus (T2DM) and atherosclerotic cardiovascular disease (ASCVD). The pathogenesis of MetS involves both genetic and acquired factors triggering oxidative stress, cellular dysfunction and systemic inflammation process mainly responsible for the pathophysiological mechanism. In recent years, MetS has gained importance due to the exponential increase in obesity worldwide. However, at present, it remains underdiagnosed and undertreated. The present review will summarize the pathogenesis of MetS and the existing pharmacological therapies currently used and focus attention on the beneficial effects of natural compounds to reduce the risk and progression of MetS. In this regard, emerging evidence suggests a potential protective role of bergamot extracts, in particular bergamot flavonoids, in the management of different features of MetS, due to their pleiotropic anti-oxidative, anti-inflammatory and lipid-lowering effects.

## 1. Introduction

Among the so called non-communicable diseases, which represent the major cause of morbidity and mortality worldwide, Metabolic syndrome (MetS) can be considered the real scourge globally [1]. It is variously known as Raven syndrome, insulin resistance syndrome, plurimetabolic syndrome and some others. MetS has been increasingly recognized as one of the important contributors to the pandemic of ASCVD and T2DM, which represent a serious clinical and public health problem [2]. Through the years, several definitions of MetS have been suggested. In the United States, the National Cholesterol Treatment Program Adult Treatment Panel III (NCEP–ATP III) published a clinically applicable definition for MetS in 2001, which was revised in 2005 establishing lower waist circumference thresholds and fasting blood glucose levels [3]. According to the NCEP ATP III definition, which is the most widely applied criteria to date, MetS is identified if three or more of the following five criteria are present: waist circumference over 40 inches (men) or 35 inches (women), blood pressure over 130/85 mmHg, fasting TG level over 150 mg/dL, fasting HDL-C level less than 40 mg/dL (men) or 50 mg/dL (women) and fasting blood sugar over 100 mg/dL (Table 1). The distinctive features of MetS included abdominal obesity, atherogenic dyslipidemia (elevated TG, small low-density lipoprotein–LDL particles, low HDL-C), elevated blood pressure, insulin resistance (with or without glucose intolerance), endothelial dysfunction and pro-thrombotic and pro-inflammatory states [4]. Often, the incidence of MetS parallels the incidence of obesity and/or T2DM. MetS is also associated with non-alcoholic fatty liver disease state (NAFLD), which has been defined as the hepatic manifestation of MetS. It was observed that MetS and NAFLD often coexist and about 90% of NAFLD patients show more than one component of MetS [5]. Indeed, some common pathophysiological features, such as increased TG, blood pressure, glucose and lower HDL levels underlying the development of NAFLD and MetS have been identified [6]. Moreover, several scientific reports have clearly recognized insulin resistance as a key factor in the pathophysiology of both diseases [5,7]. The incidence of MetS varies according to age, gender, socioeconomic status, ethnic background and criteria used for diagnosis. Several studies published in the past years reported that about one-quarter of adult manifest MetS criteria in multiple ethnic backgrounds [8]. Hence, considering the origin of Mets as a cluster of individual risk factors for disease and its spread worldwide, an accurate diagnosis to forecast the risk is crucial. The main purpose is to counteract the different components of MetS with therapeutic lifestyle changes (TLCs) and pharmacologic therapies to prevent disease, above all CVD and diabetes. In addition, there is growing interest in the use of naturally occurring compounds to reduce the risk and progression of MetS. In particular, in the last decade, interesting studies highlighted the beneficial effects of various natural antioxidants and their possible mechanisms of action for managing MetS. 

Among these, bergamot has recently been studied. Bergamot (*Citrus bergamia* Risso et Poiteau) is an endemic plant growing in Calabria (Southern Italy). Already around 1660, the essence of bergamot was used as a pain reliever and, a little later, as fragrance. Bergamot has always played an important role in Calabrian economy as the main source for the production of the essential oil used in the cosmetic industry. It has been also traditionally used in the gastronomic field and in the pre-operative natural disinfection. Bergamot possesses a profile of flavonoids and glycosides, such as neoeriocitrin, neohesperidin, naringin, rutin and poncirin, which can be considered unique in its various forms (essential oil, hydro-alcoholic extract and fruit juice), and it differs from other citrus fruits not only for the composition of its flavonoids but also for their particularly high juice content [9,10]. Recent scientific data well identified the strong anti-oxidative and anti-inflammatory effects as well as interestingly hypolipidemic and hypocholesterolemic properties shedding a new light on its use as nutraceutical.

## 2. Pathogenesis of Metabolic Syndrome

Several theories have been put forward to describe a common underlying mechanistic pathway for MetS. Whether the distinct features of MetS can be considered as distinct pathologies or are expression of a common disease is still under debate. Among the main factors contributing to the onset of MetS, lifestyle factors such as overeating and lack of physical activity have been identified. It has been suggested that visceral adiposity represents the main pathological condition that leads to the activation of most of the pathways involved in MetS [11,12]. According to this hypothesis, the high caloric intake could be considered one of the casual factors in the progression of MetS. Among the studied mechanisms involved in the development of MetS and in its transition to ASCVD and/or T2DM, insulin resistance, neurohumoral activation and chronic inflammation have been reported (Figure 1). Visceral fat has a gene expression pattern associated with higher risk of developing insulin resistance and producing smaller low-density lipoprotein cholesterol (LDL-C) size, increased LDL-C and very low–density lipoprotein (VLDL) and reduced HDL particle numbers [13]. In predisposed individuals, insulin-resistance condition leads to hyperinsulinemia, enhancing the hormone-sensitive lipase activity [14]. This condition triggers an additional lipolysis of stored TG from adipocytes raising the release of free fatty acids (FFAs) [14]. Circulating FFAs, in turn, inhibit the anti-lipolytic effect of insulin, creating a vicious cycle [15]. The increased amount of FFAs to the liver enhance their hepatic esterification to TG resulting in increased VLDL production, hypertriglyceridemia and reduction in plasma HDL-C [16]. Additional TG are further transferred to LDL, which become the main substrate for hepatic lipase resulting in more atherogenic small dense LDL (sdLDL) particles more prone to oxidation and uptake into the arterial wall [17]. With an impaired hepatic metabolism of cholesterol and excess circulating FFAs, gluconeogenesis is increased leading to hyperglycemia [18]. Several studies identified the mechanism of action underlying FFAs accumulation and muscle insulin resistance. Indeed, FFAs, converted to diacylglycerides (DAGs) and ceramides in the liver, trigger protein kinase C (PKC) activity and inhibit the protein kinase B (PKB or Akt) leading to insulin resistance [19]. This clinical condition often leads to the onset of T2DM since pancreatic β-cell function collapses trying to overcome resistance [20]. Moreover, insulin resistance is involved in the development of hypertension through different mechanisms of action. Partially, the increased circulating FFAs enhance reactive oxygen species (ROS) production, which in turn reduce the bioavailability of nitric oxide (NO) leading to increased vascular tone, vasoconstriction and elevated blood pressure [21]. Additional mechanisms involved in the development of hypertension include the effect of adipose tissue–derived cytokines [22] and hyperactivity of the renin-angiotensin-aldosterone system (RAAS) as it was observed in obese patients [23]. Visceral adipose tissue, which is considered as an active endocrine organ, synthesizes significantly high amounts of bioactive molecules, adipocytokines, which regulate inflammation process, immune function and also insulin sensitivity, blood pressure homeostasis, glucose and lipid metabolism [24]. It is well known that adipocyte products secrete monocyte chemoattractant protein-1 (MCP-1), tumor necrosis factor (TNF)-α and interleukin (IL)-6, which cause infiltration of macrophages into adipose tissue contributing to the onset of MetS [25]. In turn, TNF-α signaling activates intracellular kinases, such as c-Jun N-terminal kinase (JNK) and inhibitor of kappa B kinase (IKK), which increase serine phosphorylation of insulin receptor substrate-1 (IRS-1) impairing insulin-induced glucose uptake [26]. In an insulin resistance condition, the inhibition of Akt, due to phosphoinositide 3-kinase (PI3K) downregulation, leads to disruption of glucose transporter type 4 (GLUT-4) translocation to the surface membrane of skeletal muscle cells inhibiting glucose uptake [26]. Moreover, Akt inhibition triggers the activation of the transcription factor forkhead box protein O1 (Foxo1), which increases the expression of key enzymes of gluconeogenesis, leading to hyperglycemia [14]. Furthermore, high levels of TNF-α and IL-6 exacerbate inflammation through activation of the pro-inflammatory transcription factor nuclear factor kappa-light-chain-enhancer of activated B cells (NF-κB) [25]. NF-κB increases the release of cytokines and chemokines, the recruitment of monocytes and neutrophils to the tissues [27] and upregulates vascular cell adhesion molecules (VCAM) on endothelial cells, which leads to foam cell formation and atherosclerosis [28]. 

### Clinical Management of MetS components

The treatment of patients with MetS includes different approaches that involve lifestyle and dietary changes for weight loss and pharmacological interventions to treat atherogenic dyslipidemia and hypertension in order to decrease ASCVD and T2DM events. As described, many patients with MetS are obese because of high-calorie intake and unnecessary amount of calories. In these patients, weight loss through lifestyle modification and physical activity represent a key strategy. In a recent review, 11 randomized controlled studies of lifestyle interventions showed that the rate of patients recovered from MetS was approximately 2-fold over controls with a reduction in blood pressure, TG, waist circumference, and fasting glucose levels [29]. Dietary recommendations have been published in the American Heart Association/National Heart, Lung and Blood Institute (AHA/NHLBI) update on the NCEP criteria. According to it, fat intake should be 25% or less of calories, with reduced levels of saturated and trans fats, cholesterol intake, sodium and simple sugars [20]. A lifestyle consisting in both balanced diet and exercise is recommended as the first line treatment for MetS [30]. Weight reduction and maintenance of weight are essential and helpful in improving all the components of MetS. The Mediterranean and DASH (Dietary Approaches to Stop Hypertension) diets have demonstrated to be successful in reducing weight and improving MetS components [31,32]. Mediterranean diet, mainly based on the consumption of monounsaturated fatty acids from olives and olive oil, wholegrain cereals, fruits, vegetables, low-fat dairy, fish and nuts, has been associated with an improvement of cardiovascular outcomes decreasing oxidative stress and inflammation and improving endothelial function [33]. In addition, the *Prevención con Dieta Mediterránea* trial (Predimed) reported interesting results supporting the beneficial effects of Mediterranean diet in preventing diabetes and MetS [34]. It was shown that just an ounce of extra virgin olive oil added to the usual western type diet reduced the incidence of MetS and hypertension [34]. Interestingly, it was also observed that the molecular mechanisms underlying the beneficial effects involve the polyphenols contained in the Mediterranean diet, which cause the reduction of ROS-mediated activation of NF-κB, matrix metalloproteinases (MMPs) and cyclooxygenase-2 (COX-2) [21]. 

Beyond the application of these strategies, pharmacological interventions are used. Major traditional drugs include the use of metformin, which is, to date, the most effective and sole antidiabetic drug whose mechanism of action seems to mimic the effect of exercise [35]. Another widely used traditional drug is statin (3-hydroxy-3-methylglutaryl coenzyme A (HMG-CoA) reductase inhibitor), a potent cholesterol lowering drug with a strong effect mediated by the inhibition of the rate limiting step in the mevalonate synthesis. Statin is often used as a monotherapy or in association with other drugs targeted to decrease LDL-C levels [36]. Moreover, fibrates (a class of synthetic peroxisome proliferator-activated receptors (PPARs) agonists) are considered one of the main choice drugs in the treatment of patients with atherogenic dyslipidemia. Indeed, fibrates are effective in reducing TG levels more than 50% and increasing total HDL-C levels up to 25-30% [37]. Their different pharmacological properties have also been observed in several clinical trials carried out on patients with combined hyperlipidemia, diabetic dyslipidemia and NAFLD as well in combination therapy with statins [38,39,40]. Antiplatelet therapy is frequently used to reduce prothrombotic risk, and insulin sensitizers are administered to decrease the risk of diabetes [36]. Obese patients are treated with angiotensin converting enzyme inhibitor or angiotensin receptor blocker in presence of an increased RAAS activity [23]. To counteract salt sensitivity, the use of diuretics is acknowledged and, in some cases, recommended [23]. Although the available drugs are both safe and very effective in the treatment of the individual features of MetS, the prolonged use of more concurrent drugs could lead to serious side effects such as myopathy, pancreatitis, and thrombotic events [37,41]. However, the description of each individual therapy is beyond the scope of this review. Instead, in addition to approved pharmacological therapies, there is growing interest in the use of naturally occurring compounds as alternative strategies that could be effective in counteracting multiple components of MetS maybe avoiding the onset of side effects. Here, we describe some natural remedies useful in the management of subjects with MetS.

## 3. Natural remedies in the management of Metabolic Syndrome

Many natural remedies, which include plant extracts, spices, herbs and essential oils, have interesting therapeutic potential in the treatment of patients with MetS, which could be harnessed to create newer therapeutic modalities. Among the many existing so-called functional foods, effective in preventing MetS, berberine is of particular importance. It is a benzylisoquinoline alkaloid of the protoberberine type found in an array of plants (es. *Rhizoma coptidis*). Berberine is widely used in traditional Chinese medicine for its anti-microbial properties and has shown pharmacological biocompounds that include anti-inflammatory and antioxidant activities [42]. From twenty-seven randomized controlled clinical trials, it was reported that berberine treatment has significant therapeutic effects on patients with T2DM, hyperlipidemia and hypertension [43] In these patients, berberine reduces fasting blood glucose, postprandial plasma glucose and systolic blood pressure [43]. Furthermore, lower TG, total cholesterol (TC) and LDL-C levels were observed together with higher HDL-C levels [43]. Some recent studies have confirmed that the application of berberine effectively regulates blood glucose and lipids, ameliorates insulin resistance, inhibits inflammatory response [44] as well as reduces waist circumference, TG levels and systolic blood pressure in patients with MetS [45]. Moreover, berberine treatment results in the inhibition of human preadipocyte differentiation and leptin and adiponectin secretion. These processes are accompanied by downregulation of PPAR*γ*2, CCAAT-enhancer-binding proteins (C/EBP*α*), adiponectin and leptin mRNA expression [46]. Interestingly, Jiayu Lin et al. have shown that berberine improves metabolic function in a mouse model fed with high fat diet (HFD) [47]. Indeed, in the animals treated with berberine, an increased energy metabolism and glucose tolerance was observed. In addition, the up-regulation of thermogenesis genes (uncoupling protein 1-UCP1, and phosphor signal transducer and activator of transcription 3-p-STAT3) and the reduction of pro-inflammatory cytokines (IL-6, TNFα and MCP1) and macrophages were observed in the white adipose tissue leading to a reduction of apoptotic gene expression [47]. Berberine also exerts an insulin sensitizing action, similar to that of metformin and thiazolidinediones, in obese and diabetic C57BLKS/J-*Lepr^db^/Lepr^db^* male mice and in Wistar rats fed with HFD, which is mediated, at least in part, by AMP associated protein kinase activity [48]. One of the main drawbacks of berberine is its poor oral bioavailability, considered to be less than 1%, due to its low aqueous solubility [49]. In this regard, different pharmaceutical formulations such as microemulsion [50] and anhydrous reverse micelle system [51] have been developed to enhance its bioavailability. Despite low bioavailability and low plasma levels, high tissue distribution of berberine and its active metabolites has been observed [52]. Berberine mainly accumulates in the liver, followed by distribution in several other organs and finally in fat where it remains relatively stable for 48h [52]. Several studies performed on animal models and humans have shown that berberine undergoes demethylation in the phase I of liver metabolism followed by conjugation with glucuronic acid or sulfuric acid in phase II [53]. After oral administration, berberine and its metabolites are excreted in bile, urine and feces [54]. The safety profile of berberine has been widely studied in several human reports [55,56]. Some interactions of berberine with traditional drugs have been identified. Among others, clear synergistic effects between berberine and doxorubicin [57] or fluconazole [58] have been documented. Moreover, berberine when co-administered with L-DOPA exerts an antagonist action [59], while the concomitant use of tetradine enhances its hypoglycemic effect [60].

In recent years, another extensively studied natural antioxidant compound was curcumin [1,7-bis(4-hydroxy-3-methoxyphenyl)-1,6-heptadiene-3,5-dione]. It is the principal curcuminoid present in the *Curcuma longa*, a plant traditionally used in Asia as a natural remedy for several pathologies [61]. Much scientific evidence has been collected about curcumin including clinical trials of patients with MetS and in-depth studies on experimental models of obesity. A randomized, double-blind, placebo-controlled trial evaluated the lipid-lowering effects of curcumin in patients with MetS [62]. The study has shown that the intake of curcumin extract is associated with increased levels of HDL-C and reduced levels of LDL-C and TG. Moreover, in a subgroup of patients, it was also observed a reduced TC/HDL-C ratio [62]. Some other preliminary data have shown that a bioavailable form of curcumin ameliorates weight management in overweight people with MetS [63]. Curcumin also improves MetS in an experimental model of HFD fed rats [64]. The study demonstrated that curcumin significantly improves body mass, systolic blood pressure as well as serum levels of glucose, insulin, leptin, TC, TG, uric acid and malonildialdehyde (MDA) [64]. In addition, curcumin enhances catalase activity and strongly downregulates the expression level of TNF-α and NF-κB in hepatocytes [64]. Indeed, it is well described that curcumin inhibits the degradation of IκBα and the activation of IKK, related to NF-κB activation, leading to the suppression of inflammatory biomarkers, such as COX-2 and vascular endothelial growth factor (VEGF) in HFD fed rats [65]. Some other data have shown that curcumin downregulates the expression of different pro-inflammatory adipocytokines, such as chemokines (MCP-1, MCP-4) and interleukins (IL-1, IL-6, and IL-8), regulated by NF-κB activity, in adipose tissue isolated from obese mice [66]. Moreover, curcumin seems to be involved in the suppression of Jun NH2-terminal kinase (JNK), extracellular signal-regulated kinase1/2 (ERK1/2) and p38MAPK activities as it is shown in an in vitro model of 3T3-L1 adipocytes [67]. It was also proven that curcumin interrupts leptin signaling [68] and activates PPAR-γ in rat hepatic stellate cell growth [69], and it is able to increase adiponectin expression in a mouse model of obesity and diabetes [70]. Interestingly, curcumin also inhibits the wnt/β-catenin pathway, which is closely related to the onset of obesity [71]. In vitro and in vivo evaluation have shown that curcumin interferes to wnt/β-catenin pathway through different potential mechanisms such as downregulation of the transcription coactivator p300 [71] or inhibition of glycogen synthase kinase (GSK)-3β, which directly causes the phosphorylation of β-catenin [72]. Several animal and human studies have provided some information about the oral bioavailability of curcumin. In particular, Yang K et al. revealed that the oral bioavailability of curcumin in the plasma of rats treated with 500 mg/kg is only about 1%, suggesting that higher doses are necessary to observe some beneficial effects [73]. Interestingly, recent scientific advances in the development of new pharmaceutical formulations like liposome, polymeric nanoparticles, lipid-complexes and others have shown an increase in the bioavailability and in the activity of curcumin improving its beneficial effects [74]. Pharmacokinetics studies revealed that curcumin undergoes a biotransformation process in gut and liver producing curcumin glucuronides, sulphates or reduced molecules such as hexahydrocurcumin [75]. Intravenous injection of curcumin in mice suggests a tissue specific accumulation in particular in liver, lung, spleen and brain [76]. Curcumin metabolites, like curcumin glucuronide and curcumin sulphate, were also identified in plasma and urine of patients treated with at least 3600 milligrams of curcumin [77]. After oral administration, approximately 75% of curcumin metabolites are excreted in the feces [78]. No toxic effects were observed after oral administration of curcumin [77]. 

Among other natural remedies, Cinnamon (*Cinnamonium verum*), derived from the inner bark of many varieties of evergreen trees, is commonly used in Chinese and Indian traditional medicines [79]. Cinnamon extracts have shown anti-inflammatory and antioxidant properties as well as an interesting insulin-like activity [79]. In a randomized placebo-controlled trial, it was observed a significant improvement in fasting blood glucose, blood pressure and body composition in subjects with MetS treated with an aqueous extract of cinnamon [80]. Furthermore, a recent review, which collected 8 clinical trials that used *Cinnamomum cassia* in aqueous or powder form, reported a significant improvement of glycemic control in different animal models, in patients with pre-diabetes condition and in diabetic ones [81]. Interestingly, it was shown that cinnamon exerts its anti-diabetic activity through different molecular mechanisms including insulin receptor (IR) auto-phosphorlylation and de-phosphorylation, GLUT-4 receptor synthesis and translocation, modulation of hepatic glucose metabolism interfering with pyruvate kinase (PK) and phosphenol pyruvate carboxikinase (PEPCK) activities. This, in turn, alters the expression of PPARγ and inhibition of intestinal glucosidases [81]. The ability of Cinnamon in improving glycemic control and lipid levels was also described in an in vitro model of mouse 3T3-L1 adipocytes [82]. In particular, the study has shown a significant increase in mRNA and protein levels of IR, GLUT-4 and tristetraprolin in mouse 3T3-L1 adipocytes treated with cinnamon extract and polyphenols [82]. Recent data about the bioavailability and pharmacokinetics of cinnamaldehyde, one of the main active components derived from cinnamon, were collected. After oral administration, the bioavailability of cinnamaldehyde was approximatively 20% [83]. Pharmacokinetics studies have shown a well distribution of cinnamaldehyde and its metabolites throughout the body [84]. After oral administration, cinnamaldehyde metabolized into cinnamic acid mainly in the liver but also in stomach and small intestine [85]. Moreover, cinnamaldehyde can be converted into cinnamyl alcohol, being more susceptible to β–oxidation as their cinnamic acid derivatives [86]. It is suggested that the consumption of cinnamaldehyde should be limited to the dose related to the acceptable daily intake. Higher doses of cinnamaldehyde probably lead to toxic effects such as genotoxicity and hepatotoxicity [84]. The bioavailability and pharmacokinetics data about the other major cinnamon constituents, such as cinnamon polyphenols, cinnamic acid and eugenol have to be clarified.

Also noteworthy is capsaicin, which represents the major active constituent of chilly and constitutes, together with dihydrocapsaicin, about 90% of the capsaicinoids present in fruits belonging to the *Capsicum genus* [87]. Some epidemiologic data have shown that the consumption of foods high in capsaicin, improves metabolic and inflammatory status of adipose tissue and liver suggesting that dietary capsaicin is associated with lower incidence of obesity and/or MetS [88]. It was observed that the effects of capsaicin are mainly due to its agonist action on transient receptor potential cation channel subfamily V member 1 (TRPV1) [88]. Indeed, TRPV1 knockout mice fed a HFD become more obese and more resistant to insulin and leptin compared to the wild-type mice fed a HFD [89]. Capsaicin acts on the TRPV1 and PPARα receptors reducing metabolic dysregulation through an increase in expression levels of adiponectin and its receptor in an experimental model of obese mice [90]. Capsaicin is also able to increase liver X receptor (LXR) and pancreatic duodenal homeobox-1 (PDX-1) in the liver of streptozotocin-induced diabetic rats [91]. Both these proteins regulate glucose metabolism through modulation of GLUT-2, phosphoenolpyruvate carboxykinase and glucose 6-phosphatase expression levels, suggesting a role of capsaicin in gluconeogenesis inhibition and glycogen synthesis activation [91]. Several pharmacokinetics studies performed on experimental animal models reported that capsaicin is rapidly absorbed from the gastrointestinal lumen and detected in plasma 1h after oral administration. [92]. Chaiyasit et al. reported pharmacokinetics data obtained after oral administration of capsaicinoids in humans [93]. The results have shown capsaicin half-life in the plasma of approximatively 25 minutes and peak plasma concentration after 45 minutes. After less than 2h, no capsaicin was detected in the blood of the volunteers [93]. After oral administration, capsaicin and dihydrocapsaicin are mainly distributed in the liver and then in the kidney followed by the lung and rapidly metabolized within 24h. Most of the capsaicin is metabolized in the liver while a small amount was hydrolyzed in the small intestine [92]. Dihydrocapsaicin is metabolized to vanillyl alcohol, vanillic acid or vanillylamine as free forms or as their glucuronides and excreted in urine together with a small percentage of the unchanged form [92]. Limitations in the clinical use of capsaicin are related to its strong pungency and burning sensation. In this regard, several new pharmacological formulations have been designed such as chitosan microspheres, liposomes, nanoparticles or soft gel capsules able to bypass the release in the stomach [94,95,96]. Although several animal studies have provided evidence of the benefits of capsaicin in the treatment of MetS, less is known about the potential toxic effects. 

Some data are available regarding carnosic acid, a phenolic diterpene synthesized by plants belonging to the *Lamiaceae* family (*Rosmarinus officinalis, Salvia officinalis)*. Zhao et al. have shown that carnosic acid, the main active compound of plants, improves obesity and MetS features in an experimental model of HFD fed mice [97]. In addition to decreasing serum levels of TG, TC, insulin and glucose, carnosic acid is capable of inhibiting the expression levels of various pro-inflammatory cytokines such as IL-1β, IL-6 and TNF-α [98]. Moreover, it promotes the expression of anti-apoptotic protein B-cell lymphoma 2 (Bcl-2) and decreases the expression of pro-apoptotic protein Bcl-2-like *protein* 4 (Bax) and MMP-9 [98]. In an experimental animal study, the oral bioavailability of carnosic acid was 40%, and it was recorded 6h after administration, suggesting a slow absorption of the compound [99]. After oral administration, the main metabolites of carnosic acid such as glucuronide conjugates, carnosol and rosmanol as well as CA 12-methyl ether and 5,6,7,10- tetrahydro-7-hydroxyrosmariquinone were detected in gut, liver and plasma. It has been reported that most of the compounds persist in these tissues at significant concentrations for several hours. Carnosic acid and its metabolites are mainly excreted through the fecal route [100]. No side effects were observed both in animals treated with a single acute dose of carnosic acid and carnosol [101] and after sub-chronic consumption of the doses for 64 days suggesting a low toxicity of the extract [102] (Table 2).

## 4. Citrus Bergamia

Bergamot (*Citrus bergamia* Risso et Poiteau) is an endemic plant growing in Calabria (Southern Italy). Bergamot possesses a profile of flavonoids and glycosides, such as neoeriocitrin, neohesperidin, naringin, rutin and poncirin, which can be considered unique in its various forms (essential oil, hydro-alcoholic extract and fruit juice), and it differs from other citrus fruits not only for the composition of its flavonoids but also for their particularly high juice content [9,10]. Emerging evidence suggests a potential protective role of bergamot flavonoids in the management of different features of MetS, due to their pleiotropic anti-oxidative, anti-inflammatory and lipid-lowering effects.

### 4.1. Preparation of Bergamot Polyphenolic Fraction

Bergamot juice can be concentrated by a patented method based on a preparative size exclusion chromatography, with polystyrene gel filtration, followed by eluate exsiccation to give rise to a polyphenol-enriched powder, BPF [103]. Briefly, bergamot juice was obtained from peeled-off fruits by industrial pressing and squeezing. Then, the juice was oil fraction-depleted by stripping, clarified by ultra-filtration and loaded on polystyrene resin columns absorbing polyphenol compounds of molecular weight between 300 and 600 Da. The polyphenol fractions obtained were thus eluted by a mild KOH solution. Next, the fitocomplex was neutralized by filtration on cationic resin at acidic pH. Finally, it was vacuum dried and minced to the desired particle size to obtain BPF powder. The following HPLC analysis performed on BPF powder has shown that flavonoids are over 200 times more concentrated than those contained in bergamot juice. It was also estimated that BPF contains over 45% flavonoids, of which 95% are flavanones and 5% flavones, as well as carbohydrates, pectins and other compounds. Specifically, titration for some BPF compounds was performed showing the percentage of neoeriocitrin (>9%), naringin (>11%), neohesperidin (>11%), melitidin (>1%) and brutieridin (>2%). In addition, toxicological analyses revealed the absence of known toxic compounds including heavy metal, pesticide, phthalate and sinephrine. Moreover, no mycotoxins and bacteria were detected after standard microbiological tests. [103].

### 4.2. Lipid-lowering and Anti-diabetic Effects of BPF

In the context of MetS, several beneficial effects of BPF have been detected both in clinical trials and in experimental models. In this regard, BPF has shown important properties when administered in patients suffering from isolated hypercholesterolemia, patients with hyperlipidemia (hypercholesterolemia and hypertriglyceridemia) and patients with mixed hyperlipidemia associated with hyperglycemia [103]. All patients received an oral dose of BPF (500 mg or 1000 mg) for 30 consecutive days. At the end of the treatment period, all patients have shown a strong reduction in TG, TC, LDL-C, blood glucose levels and a significant increase in HDL-C, which is dose-dependent. Interestingly, reduced excretion level of urinary mevalonate was reported suggesting a direct inhibitory action of BPF on HMG-CoA reductase activity. The latter evidence is probably due to the structural similarity to HMG-CoA reductase substrate shown by bruteridine and melitidine, which are 3-hydroxy-3-methylglutaryl derivatives of hesperetin and naringenin, respectively. Furthermore, BPF improves the impaired endothelium-mediated vasodilation in all treated patients. The reduction of all cholesterol parameters, due to BPF treatment, was also shown in a sub-group of patients with a relevant intolerance to statins [103]. Gliozzi and colleagues well demonstrated that the co-treatment with rosuvastatin (10 mg/daily/p.o.) and BPF (1000 mg/daily/p.o.) for 30 days significantly enhances the effect of rosuvastatin alone on serum lipemic profile of patients with hyperlipemia [104]. This effect is associated with significant reduction of MDA, lectin-type oxidized LDL receptor 1 (LOX-1) and p-PKB levels, suggesting a multi-action potential for BPF in patients on statin therapy [104]. In a work published in 2014, the same research group studied the effect of BPF on LDL small dense particles and NAFLD, another important biomarker for the development of cardiometabolic risk, in patients with MetS [105]. Interestingly, a significant reduction in serum TC, LDL-C and TG was shown in patients treated with BPF (650 mg, twice a day, p.o.) for 120 consecutive days. This effect is associated with a significant reduction of serum glucose, transaminases, gamma-glutamyl-transferase and inflammatory biomarkers such as TNF-α and C-reactive protein (CRP) [105]. Moreover, BPF is able of a substantial re-arrangement of lipoprotein particles. It reduces LDL small-size atherogenic particles and enhances large-size anti-atherogenic HDL particles [105]. 

In a randomized, placebo-controlled trial, performed on MetS patients with elevated atherogenic index of plasma (AIP) and moderate hyperglycemia, the efficacy of a new bergamot juice-derived formulation was reported [106]. Bergamot polyphenolic extract complex (BPE-C) is enriched with flavonoids, pectins and vitamin C. In patients treated with 650 or 1300 mg of BPE-C for 90 consecutive days, the clear improvement of dyslipidemia was confirmed, as previously reported [106]. In addition, a powerful reduction of AIP and the amelioration of insulin sensitivity accompanied by weight loss were observed. This evidence is associated with a significant reduction of circulating leptin and ghrelin and upregulation of adiponectin [106]. Recently, BPF novel phytosomal formulation (BPF phyto) was developed to reach a better absorption and tissue distribution of BPF in patients suffering from T2DM and mixed hyperlipemia [107]. After randomization, patients received BPF (650 mg/p.o.) or BPF Phyto (500 mg/p.o.) twice a day for 30 consecutive days. The data obtained well confirmed previous results showing the beneficial effects of BPF in improving lipid profile of patients undergoing MetS with elevated cardiometabolic risk [107]. 

The lipid-lowering and anticholesterolemic effects of BPF were previously observed in rat fed with hypercholesterolemic diet [103]. The data have shown that the administration of BPF for 30 days produces a significant reduction in TG, TC and LDL-C accompanied by moderate elevation of HDL-C. Moreover, in the BPF-treated group, a better epato-biliary turnover and cholesterol consumption was observed as suggested by increased levels of total bile acids and neutral sterols in fecal samples [103]. The beneficial properties of BPF in counteracting the detrimental features of NAFLD were studied in cafeteria (CAF) diet-induced rat model of MetS [108]. The results confirmed that BPF had a role in reducing serum TG, blood glucose and obesity. Moreover, BPF counteracts hepatic steatosis strongly decreasing the amount of lipid droplets in rat hepatocytes. BPF also prevents the pathogenic lipid accumulation by stimulating the autophagic process in the liver. Specifically, the phytocomplex exerts a potent induction of lipophagy, as documented by the higher levels of LC3II found in the lipid droplet (LD) subcellular fractions of BPF-expose livers [108]. 

Moreover, a detailed work published on Scientific Report in 2020 well demonstrated that BPF is able to improve dyslipidemia and different pathophysiological features in a diet-induced mouse model of NAFLD [109]. Interestingly, the results have shown that BPF improves glucose tolerance and insulin resistance as well as liver enzymes and dyslipidemia counteracting non-alcoholic steato-hepatitis (NASH) [109]. BPF is able to reduce oxidative stress markers along with JNK and p38 MAP kinase activity. BPF also prevents the exacerbation of inflammatory process and sinusoidal fibrosis in the liver [109]. In an in-depth study, a molecular mechanism partly responsible for the hypolipemic properties of BPF was also clarified [110]. In vivo data demonstrated that BPF prevents alteration of lipid profile in rats fed with hypercholesterolemic diet, reducing oxidative stress and ameliorating lipoprotein metabolism dysregulation [110]. This, in turn, restores the activity of acetyl-coenzyme A acetyltransferase (ACAT), lecithin cholesterol acyltransferase (LCAT), cholesteryl ester transfer protein (CETP) and paraxonoase-1 (PON1), an effect accompanied by the concomitant normalization of apolipoprotein A1 (Apo A1) and apolipoprotein B (Apo B) levels [110]. 

### 4.3. Antioxidant and Anti-Inflammatory Effects of BPF

The antioxidant effect of bergamot was initially observed by analyzing the potential activity of the non-volatile fraction of the bergamot essential oil (BEO-NVF) in an experimental model of neointima hyperplasia. Mollace et al. identified the highly significant effect of the antioxidant component of BEO-NVF on LOX-1 expression and ROS generation in a model of rat carotid artery injury [111]. The results have shown that balloon injury is associated with smooth muscle cell proliferation and neo-intima formation causing re-stenosis. These cells also reveal an increase in LOX-1 expression and generation of ROS. Interestingly, pre-treatment of rats with BEO-NVF decreases neo-intima formation, LOX-1 expression and ROS generation as well as the degree of stenosis [111]. The natural antioxidant and LOX-1 modulating properties of BEO-NVF appear to be promising for use in MetS to decrease endothelial dysfunction, smooth muscle cell proliferation and inflammation, all of which bridge the gap between MetS and ASCVD. Furthermore, BPF has shown robust antioxidant properties in a CAF diet-induced rat model of MetS ameliorating the plasmatic oxidative balance. In particular, data have shown an increase in the expression level of the antioxidant enzyme glutatione S-tranferasi P1 (GSTP1) and the inhibition of the pro-apoptotic markers caspase 8 and 9 [112]. Important evidence on the anti-inflammatory activity of BPF were collected in CAF diet-induced NASH rats. [113]. The study demonstrated the ability of BPF supplementation to decrease hepatic inflammation by reducing IL-6 and increasing anti-inflammatory IL-10 mRNA expression levels. These results correlate with fewer Kupffer cells, leukocytes infiltrating perivascular hepatic tissue and lower inflammatory foci score in CAF fed rats treated with BPF [113]. The reliable antioxidant and anti-inflammatory properties of bergamot polyphenols have also been studied in association with cynaropicrin, a sesquiterpene lactone of a guaianolide type isolated from artichoke (*Cynara cardunculus)* [114]. The phytocomplex, called Bergacyn, has been previously titrated in polyphenols derived from bergamot and artichoke. The percentage of titrated polyphenol coming from *Citrus bergamia* was > 19.5% while the one derived from *Cynara cardunculus* was >10%. Moreover, some compounds of Bergacyn have been titrated showing the percentage of neoeriocitrin (>4.5%), naringin (>5.5%), neohesperidin (>5.5%), melitidin (>1) and brutieridin (>2%). Musolino V. et al. well demonstrated the synergistic effect of Bergacyn on vascular inflammation and oxidative stress in a randomized, double blind, placebo controlled clinical study of patients with T2DM and NAFLD [114]. After 16 weeks of treatment with Bergacyn (300 mg/daily/p.o.), a significant improvement of NAFLD biomarkers (alanine aminotransferase-ALT, aspartate aminotransferase-AST, gamma-glutamyl transferase-γ-GT and alkaline phosphatase-ALP) as well as liver fibrosis biomarkers (hyaluronic acid-HA, type III precollagen-PC III and type IV collagen-IV-C) was shown in patients with T2DM [114]. These effects are associated with a substantial modulation of oxidative stress and inflammatory biomarkers. Indeed, an increase in glutathione peroxidase (GPx) and superoxide dismutase (SOD) levels and a reduction of MDA and TNF-α levels was shown [71]. Moreover, at the end of the experimental period, an improvement in endothelial dysfunction was observed, contributing to better NO-mediated reactive vasodilation [114] (Table 2, Figure 2). 

### 4.4. Bioavailability and Pharmacokinetics of BPF

Up to date, the oral bioavailability and pharmacokinetic of BPF has not been completely investigated. In the recent study of Musolino et al., oral bioavailability information of BPF has been collected. After 11 weeks of treatment, neoeriocitrin, naringin and neohesperidin have been detected in the serum of mice that received BPF by gavage, suggesting that those biocompounds are able to cross small intestine membrane [109]. Moreover, a recent pilot bioavailability study clarified how these compounds are modified after oral consumption of bergamot juice in healthy volunteers’ biological fluids [115]. The authors demonstrated that flavonoids from *Citrus bergamia* undergo phase II conjugates metabolism (such as sulfate and glucuronides of hesperetin, naringenin and eriodyctiol). They were detected at both 1h and 4h in the plasma samples of all the volunteers. The majority of conjugates were also detected in urine at 2h and 6h, accounting for the absorption profile observed in plasma [115]. Although it is unclear whether flavonoid metabolites retain some biological activity, their involvement in the modulation of intracellular responses is conceivable as has been observed for the sulfonated and methylated metabolites of resveratrol which retain, at least in part, the activity of the parent compound [116]. Toxicological studies revealing that BPF are absolutely safe, having been carried out according to EU Directive 2004/9/EC and Directive 2004/9/EC for Good Laboratory Practice Guidelines (GLP) as well as OECD Guidelines for Repeated Dose 28- and 90-day Oral Toxicity Study in Rodents [105]. In addition, comparable doses of bergacyn were used as reference protocol in terms of efficacy and safety profile in a study performed on diabetic patients [103,114].

### 4.5. Hypothesis on BPF Mechanisms of Action

The studies discussed above reveal the interesting pleiotropic effects of BPF due to its peculiar composition and the highest content of Citrus flavonoids. The molecular mechanisms underlying these effects are not fully known. However, the results obtained allow to identify some molecular mechanisms affected by BPF action.

The hypolipemic and anti-atherogenic effects of BPF are, at least in part, associated with the modulation of the activity of some enzymes responsible for cholesterol esterification reactions and lipid trafficking [110]. The improvement of lipoprotein metabolism is probably due to the restored activity of ACAT, LCAT, CETP and PON1 enzymes by BPF [110]. These enzymes differently contribute to the modification of plasma lipoprotein particle composition mediating cholesterol esterification within the cells (ACAT) [117] or in the plasma (LCAT) [118], lipid transfer in plasma compartment (CETP) [119] or through hydrolysis of lipid peroxides on LDL and HDL particles (PON1) [120]. BPF also interferes with the autophagic pathway preventing the pathogenic lipid accumulation [108]. Indeed, BPF strongly enhances autophagy in the liver upregulating the expression level of Beclin-1 and LC3II and reducing p62. Moreover, a potent induction of lipophagy, documented by the higher levels of LC3II, was found in the LD subcellular fractions of BPF-expose livers [108]. More recently, a direct action of BPF on MAPK, which represent a crucial regulator of glucose and fatty acids metabolism in all tissues, has been identified [109]. In particular, in pathological fatty liver, BPF induced modulation of JNK/p38 MAPKs represents the protective mechanism responsible for the amelioration of insulin sensitivity [109]. Furthermore, the reduction of liver inflammation and fibrosis is related to BPF direct inhibition of poly [ADP-ribose] polymerase 1 (PARP1), considered the direct suppressor of MAPK inhibitor mitogen-activated protein kinase-1 (MPK-1) [109]. It was also assumed that the reduction of serum and tissue cholesterol levels is due to the statin-like activity exerted by BPF. The structural similarity of bergamot polyphenols to HMG-CoA reductase substrate allows them to mimic the natural substrates of HMG-CoA and block the rate-limiting step in cholesterol synthesis [103]. The powerful antioxidant effects of BPF play a significant role in all the observed protective effects. BPF directly reduces lipid peroxidation biomarkers (TBARS) and MDA and strongly inhibits protein tyrosine nitration levels [104]. BPF also improves the activity of endogenous antioxidant enzymes such as SOD, GPx and GSTP1 [112,114]. The additive vaso-protective effect of BPF is closely related to its antioxidant property. BPF is able to reduce LOX-1 expression levels highly modulated in the development and progression of endothelial dysfunction to atherosclerosis [104]. The vaso-protective action of BPF is also associated with the increase in PKB phosphorylation providing protection against vascular atherogenic injury [104]. 

All together, these data add new insights into the beneficial role of bergamot and highlights the potential use of supplementation treatments of bergamot-extracts for reducing cardiometabolic disorders in patients with MetS. However, future research will be aimed to better clarify the molecular mechanisms underlying the several health effects described in this review. Moreover, although it is clear that several polyphenols contribute to the beneficial effects of BPF, the specific contribution of each of them still remains to be clarified. Furthermore, it would be of interest to assess the potential beneficial properties of other herbal compounds with BPF, which may enhance its effects. The identification of phytocomplexes with synergistic action can be useful in the management of different pathological features. 

## 5. Conclusions

MetS represents a complex pathophysiologic condition whose main determinants are central adiposity, hypertension, high TG, low HDL-C and hyperglycemia. All these components are linked by a common mechanism of development of chronic inflammation that often leads to insulin-resistance. The overlap of multiple risk factors, in each disease state, increases the risk of ASCVD and development of T2DM. To date, the management of MetS components provides lifestyle recommendations including exercise, weight loss and Mediterranean diet consumption, as well as the use of traditional pharmacologic therapies. However, the use of the approved drugs is limited by various factors such as the onset of side effects due to prolonged treatment. Instead, there is growing interest in the use of naturally occurring compounds to reduce the risk and progression of MetS. This review collected some experimental evidence about the role of nutraceuticals in the management of the different components of MetS. Preclinical and clinical studies suggest that different natural compounds have beneficial effects against obesity and insulin resistance as well as against various complications resulting from these diseases. Interestingly, it has been shown how many nutraceuticals reduce serum level of glucose, insulin, TC, TG, normalize blood pressure and are capable of a substantial re-arrangement of lipoprotein particles ameliorating serum lipemic profile. Moreover, strong antioxidant and anti-inflammatory effects as well as anti-apoptotic effects have been demonstrated. Some of the studies also identified specific activity of natural compounds in glucose metabolism, adipogenesis and their interaction with wnt/β-catenin pathway, which is closely related to obesity. In particular, bergamot and artichoke extracts, in addition to the strong anti-inflammatory, antioxidant, hypolipidemic and hypocholesterolemic properties, have shown interesting effects against the key pathophysiological features of NAFLD.

These intriguing data shed new light on the potential use of nutraceuticals for preventing and/or reducing cardiometabolic risk in patients with MetS. However, more research, in particular clinical trials, is needed to further understand the link between natural compounds and chronic diseases and the molecular mechanisms underlying their beneficial effects.

## Figures and Tables

**Figure 1 nutrients-12-01504-f001:**
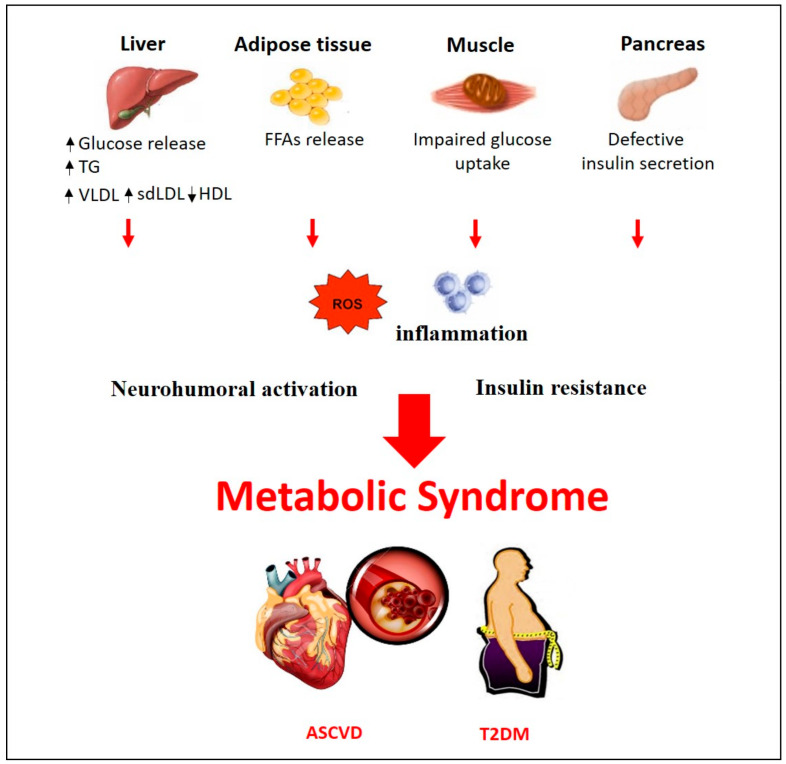
Pathogenesis of Metabolic syndrome. Pathophysiological mechanisms in metabolic syndrome. ASCVD, Atherosclerotic Cardiovascular Disease; FFAs, Free Fatty Acids; HDL, High Density Lipoprotein; sdLDL, small dense low-density Lipoprotein; TG, Triglycerides; T2DM, Type 2 Diabetes Mellitus; VLDL, very-low-density Lipoprotein.

**Figure 2 nutrients-12-01504-f002:**
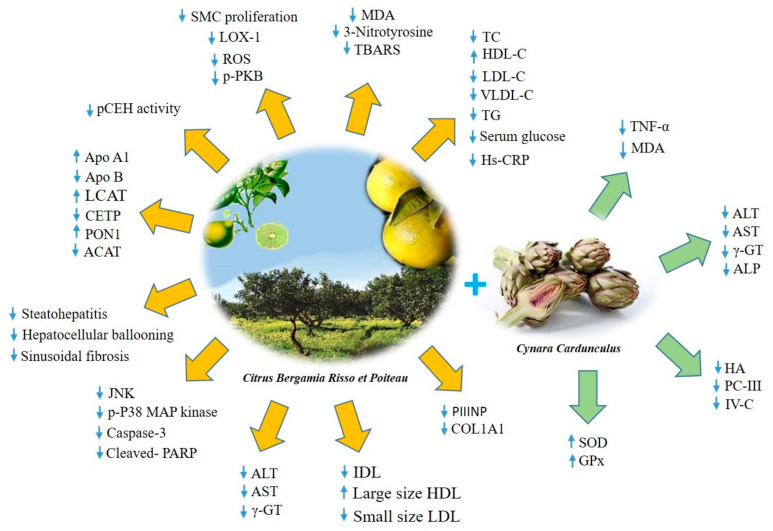
The molecular mechanisms involved in the beneficial effects of bergamot polyphenols. ACAT, Acetyl-Coenzyme A Acetyltransferase; ALP, Alkaline Phosphatase; ALT, Alanine Aminotransferase; Apo A1, Apolipoprotein A1; Apo B, Apolipoprotein B; AST, Aspartate Aminotransferase; CETP: Cholesteryl Ester Transfer Protein; COL1A1, Pro-Collagen type I; GPx, Glutathione Peroxidase; HA, hyaluronic acid; HDL-C, High Density Lipoprotein Cholesterol; Hs-CRP, High-sensitivity C-Reactive Protein; IDL, Intermediate Density Lipoprotein; JNK, c-Jun N-terminal Kinase; LCAT, Lecithin Cholesterol Acyltransferase; LDL-C, Low Density Lipoprotein Cholesterol; LOX-1, Lectin-type Oxidized LDL receptor 1; MDA, Malonildialdehyde; PARP, Poly(ADP-ribose) Polymerase; pCEH, pancreatic Cholesterol Ester Hydrolase; PC III, Pre-collagen type III; p-PKB, phospho- Protein Kinase B; PON1, Paraxonoase-1; PIIINP, Pro-Collagen III N-terminal propeptide; ROS, Reactive Oxygen Species; SMC, Smooth Muscle Cell; SOD, Superoxide Dismutase; TBARS, Thiobarbituric Acid Reactive Substances; TC, total Cholesterol; TG, Triglycerides; TNF-α, Tumor Necrosis Factor-alpha; VLDL-C, Very-Low-Density Lipoprotein Cholesterol; γ-GT, Gamma-Glutamyl Transferase; IV-C, Collagen type IV.

**Table 1 nutrients-12-01504-t001:** Diagnostic criteria for metabolic syndrome.

Clinical Measure	WHO 1998	EGIR 1999	ATP III 2001	IDF 2005	AHA/NHLBI 2005
**Criteria**	Insulin Resistance + any other 2	Insulin Resistance + any other 2	Any other 3 of 5	Increased WC (population specific) + any other 2	Any other 3 of 5
**Insulin Resistance**	IGT/IFG/IR	Plasma insulin < 75^th^ percentile	-	-	-
**Blood Glucose**	IFG/IGT/T2DM	IFG/IGT (excludes diabetes)	≥ 110 mg/gL (includes diabetes)	≥ 100 mg/gL	≥ 100 mg/gL (includes diabetes)
**Dyslipidemia**	TG ≥ 150 mg/dL HDL-C Men < 35mg/dL Women < 39mg/dL	TG ≥ 150 mg/dL HDL-C < 39mg/dL	TG ≥ 150 mg/dL HDL-C Men < 40 mg/dL Women < 50 mg/dL	TG ≥ 150 mg/dL or on TG treatment HDL-C Men < 40 mg/dL Women < 50 mg/dL or HDL treatment	TG ≥ 150 mg/dL or on TG treatment HDL-C Men < 40 mg/dL Women < 50 mg/dL or HDL treatment
**Blood Pressure**	≥ 140/90 mmHg	≥ 140/90 mmHg or on treatment	≥ 130/85 mmHg or on treatment	≥ 130/85 mmHg or on treatment	≥ 130/85 mmHg or on treatment
**Obesity**	Waist:Hip (W:H) ratio Man > 0,9 Women > 0,85 and/or BMI > 30 Kg/m^2^	WC Men ≥ 94 cm Women ≥ 80 cm	WC Men ≥ 102 cm Women ≥ 88 cm	WC ≥ 94 cm	WC Men ≥ 102 cm Women ≥ 88 cm
**Other**	Microalbuminuria	-	-	-	-

ATP, Adult Treatment Panel; BMI, Body Mass Index; EGIR, European Group for Study of Insulin Resistance; HDL-C, High Density Lipoprotein Cholesterol; IDF, International Diabetes Federation; IFG, Impaired Fasting Glucose; IGT, Impaired Glucose Tolerance; IR, Insulin Resistance; TG, Triglycerides; T2DM, Type 2 Diabetes Mellitus; WC, Waist Circumference; WHO, World Health Organization; -: no reference values have been reported.

**Table 2 nutrients-12-01504-t002:** The main properties of different natural compounds.

Plant	Bioactive Component	Properties	In vitro/in vivo Models	Clinical Trials	References
*Rosmarinus officinalis* *Salvia officinalis*	Carnosic acid	↓ Body weight ↑ Insulin sensitivity ↓ Serum Glucose, TG, TC ↓ ALT, AST ↓ MDA, IL-1β, IL-6, TNF-α ↑ Bcl-2 ↓ Bax, MMP-9	- HFD fed mice		[34,35]
*Cinnamonium verum* *Cinnamomum cassia*	Cinnamaldehyde Polyphenols	Anti-inflammatory and antioxidant effects Insulin-like activity ↓ Fasting blood glucose and blood pressure ↑ IRβ, GLUT-4, TTP, GLP-1, PPAR-γ	- mouse 3T3-L1 adipocytes - High Fructose Diet fed mice - STZ-induced diabetic rats	- Pre-diabetes - MetS - T2DM	[36,37,38,39]
*Capsicum genus*	Capsaicin	↓ Fasting glucose ↑ Insulin sensitivity ↓ TG, Leptin ↑ Adiponectin ↓ Gluconeogenesis ↑ Glycogen synthesis ↓ TNF-α, MCP-1, IL-6 ↑ LXR, PDX-1 ↑ TRPV-1, GLUT-4, IRS-1 ↑ PPAR-α/PGC-1^α^	- TRPV1-KO mice fed with HFD - HFD fed mice - STZ-induced diabetic rats		[40,41,42,43,44]
*Curcuma longa*	Polyphenols	Anti-inflammatory and antioxidant effects ↑ Insulin sensitivity ↓ BMI, body fat, systolic blood pressure ↓ Plasma glucose ↓ NF-κB, COX-2, VEGF ↓ MCP-1, MCP-4, ILs, TNF-α ↓ JNK, ERK1/2, P38MAPK ↓ Wnt/β-catenin pathway ↓ TG, TC, Leptin ↑ Adiponectin ↓ Malondhyaldeide ↑ PPAR-γ, Catalase activity	- mouse 3T3-L1 adipocytes - rat hepatic stellate cells - HFD fed mice - ob/ob C57BL/6J mice - Balb/c mice - HFD fed rats - STZ-induced diabetic rats fed with HFD	- MetS	[45,46,47,48,49,50,51,52,53,54,55,56]
*Rhizoma Coptidis*	Berberine	Anti-inflammatory and antioxidant effects ↑ Insulin sensitivity ↓ Fasting glucose Plasma glucose, systolic blood pressure ↓ TG, TC, LDL-C ↑ HDL-C ↓ Leptin, adiponectin ↓ hs-CRP, IL-6, TNF-α, MCP-1 ↓ Macrophage recruitment ↑ Thermogenesis ↓ PPARγ2, C/EBPα, ↑ AMPK and GLUT-4	- mouse 3T3-L1 adipocytes - rat L6 myotubes - HFD fed mice - C57BLKS/J Lepr^db^-Lepr^db^ mice - HFD fed rats	- T2DM -Hyperlipemia -Hypertension - MetS	[57,58,59,60,61,62,63]
*Citrus bergamia* Risso et Poiteau	BEO-NVF BPF	↓ SMC proliferation, LOX-1, p-PKB ↓ ROS, TBARS, MDA, Nitrotyrosine ↓ Serum glucose, TG, TC, LDL-C, VLDL-C ↑ HDL-C Re-arrangement of lipoprotein particles ↓ ALT, AST, γ-GT, ↓ Hs-CRP, TNF-α, JNK, p-P38 MAPK, ↓ Caspase-3, Cleaved- PARP ↑ Lipid transfer protein system ↓ Fibrogenic activity ↓ pCEH↓ Steatohepatitis, hepatocellular ballooning ↓ Sinusoidal fibrosis	- rat neointimal hyperplasia - hypercholesterolemic diet fed rats - NAFLD mice	-Hyperlipemia - MetS - NAFLD - T2DM	[66,67,68,69,70,71,72]
*Cynara cardunculus*	Cynaropicrin	↓ TNF-α, MDA ↓ ALT, AST, γ-GT, ALP ↓ Liver fibrosis ↑ SOD, GPx		- NAFLD - T2DM	[73]
Brassicaceae family Gramineae family	Coenzyme Q10	Antioxidant capacity, nephroprotective effect ↓ TG, TC, LDL-C, serum insulin ↑ β-cell Function ↑ Glucose metabolism	- db/db and dbH mice model of type 2 diabetic nephropathy - STZ-nicotinamide induced diabetic rats	- T2DM - MetS	Zozina V. I. et al. 2018 [74]
*Vitis vinifera*	Resveratrol	↓ BMI, waist circumference, insulin secretion ↓ Hs-CRP, TNF-α ↓ Malondhyaldeide ↓ Leptin, RAAS modulation ↓ Lipogenesis ↑ Lipolysis	- SGBS preadipocytes - human preadipocytes - adipose stem cells - high Fructose Diet fed rats - high Sucrose Diet fed rats - high–fat/cholesterol diet fed swine - IH-induced metabolic dysfunction in mice - insulin-resistant KKA^y^ mice	- MetS	Hou C.Y. et al. 2019 [75]
*Vaccinium myrtillus* *Fragaria ananassa*	Anthocyanins	Anti-inflammatory and antioxidant effects Hypocholesterolemic effects ↓ TG, body weight, fat mass ↓ α-amylase and α-glucosidase activities ↓ Leptin ↓ MCP-1, ICAM-1, VCAM-1, NF-κB B ↑ PPARs	- HK-2 cells - HUVEC cells - STZ- induced diabetic rats - STZ- induced diabetic mice - db/db mice fed with HFD - obese Zucker rats - Dahl Salt-Sensitive rats	- MetS	Naseri R. et al. 2018 [76]
Oleaceae family (Olea europaea Linn.)	Oleuropein	Antioxidant effect ↑ Insulin sensitivity, glucose tolerance ↓ TG, TC, LDL-C ↑ SOD, GPx ↓ LpL, PPARγ, C/EBPα, SREBP-1c ↓ Leptin ↑ AMPK UCP-1 TRPV-1	- MSC from human bone marrow - 3T3-L1 adipocytes - C2C12 cells - Alloxan-induced diabetic rats - HFD fed mice and rats - PPARα null mice - BPA-induced hyperlipidemia and liver injury in rats	-Hyperchole sterolemia - Overweight	Ahamad J.et al. 2019 [77]

↑: Increased, ↓: Decreased. ALP, Alkaline Phosphatase; ALT, Alanine Aminotransferase; AMPK, 5’ AMP-activated Protein Kinase; AST, Aspartate Aminotransferase; Bax, Bcl-2-like *protein* 4; Bcl-2, B-cell lymphoma 2; BEO-NVF, Non-Volatile Fraction of the Bergamot Essential Oil; BMI, Body Mass Index; BPA, Bisphenol A; BPF, Bergamot Polyphenolic Fraction; C/EBP*α*, CCAAT-Enhancer-Binding Protein-*α*; COX-2, Cyclooxygenase-2; ERK1/2, Extracellular signal-Regulated Kinase 1/2; GLP-1, Glucagon-Like Peptide-1; GLUT-4, Glucose Transporter Type 4; GPx, Glutathione Peroxidase; GSK-3β; Glycogen Synthase Kinase-3β; HDL-C, High Density Lipoprotein Cholesterol; HFD, High Fat Diet; Hs-CRP, High sensitivity C-Reactive Protein; ICAM-1, Intercellular Adhesion Molecule 1;IH, Intermittent Hypoxia; IL-1β (Interleukin-1β); IL-6, Interleukin.6; IRS-1, insulin receptor substrate-1; IRβ, Insulin Receptor- β; JNK, c-Jun N-terminal Kinase; LDL-C, Low Density Lipoprotein Cholesterol; LOX-1, Lectin-type Oxidized LDL receptor 1; LpL, Lipoprotein Lipase; LXR, Liver X Receptor; MCP-1, Monocyte Chemoattractant Protein-1; MDA, Malonildialdehyde; MetS, Metabolic Syndrome; MM-9, Matrix Metalloproteinase-9; NAFLD, Non Alcoholic Fatty Liver Disease; NF-κB, nuclear factor kappa-light-chain-enhancer of activated B cells; PARP, Poly(ADP-ribose) Polymerase; pCEH, pancreatic Cholesterol Ester Hydrolase; PDX-1, Pancreatic Duodenal homeobox-1; PGC-1α, PPAR-γ coactivator-1α; PPAR-γ, Peroxisome Proliferator-Activated Receptor- *γ*; p-PKB, phospho-Protein Kinase B; p-38 MAPK, p-38 Mitogen-Activated Protein Kinases; RAAS, Renin Angiotensin Aldosterone System; ROS, Reactive Oxygen Species; SMC, Smooth Muscle Cell; SOD, Superoxide Dismutase; SREBP-1c, Sterol Regulatory Element-Binding Protein-1c; STZ, Streptozotocin; TBARS, Thiobarbituric Acid Reactive Substances; TC, total Cholesterol; TG, Triglycerides; TNF-α, Tumor Necrosis Factor-alpha; TRPV-1, Transient Receptor Potential cation channel subfamily V member 1; TTP, Tristetraprolin; T2DM, Type 2 Diabetes Mellitus; UCP-1, Uncoupling Protein-1; VCAM-1, Vascular cell adhesion protein 1; VEGF, Vascular Endothelial Growth Factor; VLDL-C, Very-Low-Density Lipoprotein Cholesterol; γ-GT, Gamma-Glutamyl Transferase.

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
