# Peer review of "The Effect of Natural Antioxidants in the Development of Metabolic Syndrome: Focus on Bergamot Polyphenolic Fraction"

_nutrients, 2020, doi:10.3390/nu12051504_

Round 1

Reviewer 1 Report

The review is focused on the possible role of natural antioxidants in the management of the metabolic syndrome.

Main comments:

  • Extensive revision and improvement of English language is needed.
  • The link between the metabolic syndrome and NAFLD is well established and should be discussed in the introduction.
  • Information on the various natural products should be listed in order of evidence, starting with those having evidence in humans. When available, please add information on oral bioavailability and kinetic parameters, known interactions and contraindications.
  • The level of evidence should be added to table 2 (in vitro studies/animal models/clinical trials).
  • The use of standard abbreviation is recommended (for example TC and not t-chol for total cholesterol). Please define TLC (line 52).
  • Page 5: regarding pharmacological therapies for MetS-associated dyslipidemia, fibrates should be also mentioned, due to their ability to decrease TG and increase HDL-C. In addition, I do not agree with the statement at lines 162-163 “Unfortunately, the drugs that actually have significant effects are few and often their prolonged use leads to serious side effects”. Available drugs for the treatment of dyslipidemia, insulin resistance/diabetes, hypertension, etc. are very effective against the individual features of MetS and their safety profile is good. Please rephrase.

Reviewer 2 Report

The manuscript presented is well redacted in a grammatical term, however very dense for its comprehension for the readers. One big problem is identified in this review, which is the lack of scope and objective. In the title of the manuscript, it is mentioned that it will be focused on the Bergamot Polyphenolic Fraction; however, until page 9 of 12 written pages, no special focus has been done to Bergamot. This is also observed in the abstract, which includes well-know features of metabolic syndrome, without mentioning the described mechanisms of action of the Bergamot polyphenolic fraction. Similarly, in the conclusion, the reader will appreciate that the authors could somehow determine from all the information collected from several studies which are the main pathways in which Bergamot polyphenolic fraction could be useful and in which specific circumstances. And finally, to determine the perspective of future studies to validate the use of such a class of compounds.

My suggestion is to reduce all the description dedicated to the pathology of metabolic syndrome, and focus on specific features of Bergamot:

  • What is Bergamot, its traditional consumption
  • Detailed analysis of Bergamot polyphenolic fraction.
  • Describe separately (e.g. inflammation, lipid metabolism, oxidative stress, etc) in different heading the knowledge of the effects of Bergamot to counteracts these symptoms 
  • Identified the main mechanism of action from all the information described (this will the author's main contribution to the field) and the future fields of investigation.
  • Similarly, it will be interesting to describe the dose of action, possible side-effects, etc.

Round 2

Reviewer 1 Report

The authors modified the manuscript according to the reviewers’ comments and suggestions.

Additional comments:

  • Paragraph 4.2. “Lipid-lowering” and “anti-cholesterolemic” sound like a repeat. Lipid-lowering is likely more appropriate due to the action on TG and mixed dyslipidemias. When referring to lipoproteins, please use LDL particles and not LDL-C particles (the same for HDL).
  • Paragraph 4.5. ACAT, LCAT, CETP and PON are involved in lipid/lipoprotein metabolism; their effect is complex and not limited to cholesterol. ACAT and LCAT do not directly mediate lipid transfer, but cholesterol esterification within the cells (ACAT) or in the plasma compartment (LCAT). CETP is a lipid transfer protein, as many others in plasma compartment (for example PLTP) or in the cells (for example MTTP). PON is an antioxidant enzyme mainly acting on phospholipids. The definition of the “Lipid Transfer Protein System” as described here is not supported by the literature in the field of lipidology. What is the meaning of “lipo-lipoprotein particles”? Lines 672-677 must be rephrased. The authors could speculate that bergamot modulates lipid levels by affecting lipoprotein metabolism, as indicated by the data on ACAT, LCAT, etc.
  • Please check again the text for English language. There are still some mistakes and misspelling (for example “to rich” instead of “to reach”, single/plural form of verbs, etc).
  • In the legends, please list the abbreviations in alphabetical order.

Reviewer 2 Report

The authors included all the suggestions raised by the reviewers. I don't have additional comments. 

Author Response

We thank Reviewer #2 for her/his overall appreciation for the work submitted in the first round of revision.

This manuscript is a resubmission of an earlier submission. The following is a list of the peer review reports and author responses from that submission.